# Direct molecular level characterization of different heterogeneous freezing modes on mica – Part 1

Ahmed Abdelmonem[1]

[1]Institute of Meteorology and Climate Research – Atmospheric Aerosol Research (IMKAAF), Karlsruhe Institute of Technology (KIT), 76344 Eggenstein-Leopoldshafen, Germany

*Correspondence to*: Ahmed Abdelmonem (ahmed.abdelmonem@kit.edu)

**Abstract.** The mechanisms behind heterogeneous ice nucleation are of fundamental importance to the prediction of the occurrence and properties of many cloud types, which influence climate and precipitation. Aerosol particles act as cloud condensation and freezing nuclei. The surface–water interaction of an ice nucleation particle plays a major, not well explored, role in its ice nucleation ability. This paper presents a real–time–molecular–level comparison of different freezing modes on the surface of an atmospherically relevant mineral surface (mica) under varying supersaturation conditions using second harmonic generation spectroscopy. Two sub-deposition nucleation modes were identified (one- and two-stage freezing). The nonlinear signal at water-mica interface was found to drop upon the formation of a thin film on the surface regardless of 1) the formed phase (liquid or ice) and 2) the freezing path (one– or two–step), indicating similar molecular structuring. The results also revealed a transient phase of ice at water–mica interfaces during freezing, which has a lifetime of around one minute. Such information will have a significant impact on climate change, weather modification, and tracing of water in hydrosphere studies.

## 1 Introduction

Clouds influence the energy budget by scattering sunlight and absorbing heat radiation from the earth and are therefore considered the major player in the climate system. Formation of ice changes cloud dynamics and microphysics because of the release of latent heat and the Bergeron-Findeisen process, respectively (Pruppacher and Klett, 1997). Ice nucleation in the atmosphere can be triggered heterogeneously by aerosol particles, ice-nucleating particles (INP), or occurs homogeneously at about -38°C (Pruppacher and Klett, 1997). Cloud evolution depends not only on temperature and humidity, but also on the abundance and surface characteristics of atmospheric aerosols. Understanding the factors that influence ice formation within clouds is a major unsolved and pressing problem in our understanding of climate (Slater et al., 2016). Field and laboratory experiments on cloud formation started decades ago (see (Schaefer, 1949; DeMott et al., 2011; Hoose and Mohler, 2012) and references therein) and are ongoing. A wide variety of results and observations has been obtained in cloud microphysics, especially with respect to the ice nucleation ability of atmospheric aerosol particles and, hence, the mechanisms of cloud dynamics, precipitation formation, and interaction with incoming and outgoing radiation. Aerosol particles act as cloud condensation nuclei for liquid clouds, immersion or contact freezing nuclei for mixed-phase clouds, and heterogeneous deposition nuclei for ice (cirrus) clouds. Depending on whether water nucleates ice from the vapor or the supercooled liquid phase, ice nucleation is classified as deposition nucleation or immersion nucleation, respectively. Despite numerous investigations aimed at characterizing the effect of particle size and surface properties of the INP, there is a lack of information about the restructuring of water molecules on the surface of INPs around the heterogeneous freezing point.

In this paper, discrimination between different modes of freezing of water on an ice-nucleating surface using nonlinear optical spectroscopy is demonstrated. Mica, a widely spread layered clay mineral and one of the most prominent mineral

surfaces due to its atomic flatness and chemical inertness produced by perfect cleavage parallel to the 001 planes (Poppa and Elliot, 1971), was selected as a model surface in this study. However, the image of an inert and atomically smooth surface prepared by cleavage of muscovite mica in an ambient atmosphere is not quite correct (Christenson and Thomson, 2016). Surface analytical techniques found that the surface of muscovite mica cleaved in laboratory air, as the case in this work, contains a water-soluble compound, almost potassium carbonate crystallites, which may cover few tenths of a percent of the surface area (Christenson and Israelachvili, 1987). Nevertheless, definitive chemical analysis showing its presence is not yet available. However, the mobility of the potassium ions as potassium carbonate does not necessarily affect significantly any measurement of average surface properties as in the case of this work, and at high humidities the potassium will be widely dispersed across the surface. The readers which are interested in more details on the nature of the mica surface in general and air-cleaved mica surface in particular are referred to the review paper of Christenson, H. K. and Thomson, 2016 and papers cited therein.

Mica, as natural particles, is believed to be among the most effective ice nucleating minerals in the deposition mode (Eastwood et al., 2008; Mason and Maybank, 1958). An early study, using a projection microscope, of deposition nucleation of ice on freshly cleaved synthetic fluorophlogopite mica, which is similar in structure to muscovite but has the hydroxyl groups replaced by fluorine, revealed that there is no growth of ice until saturation with respect to water is reached (Layton and Harris, 1963). The authors concluded that at temperatures above -40 °C, the growth of ice on mica should be a two-step process: A nucleus forms as water and then freezes. Experimental evidence of two-step nucleation was also provided by (Campbell et al., 2013) using an optical microscope. With the help of a scanning optical microscope, they showed that the nucleation, of various organic liquids crystallising from vapour on mica surfaces, favored specific nucleation sites with surface features such as cleavage steps, cracks and pockets. However, they suggested that a supercooled liquid phase forms first and then freezes after it has grown to a size which thermodynamically favors the solid phase. These assumptions were based merely on thermodynamic observations (temperature and vapor supersaturation). A later study by the same group, has confirmed the role of the surface features and the two-step process for the organic liquids, and strongly suggested a two-step process for water and ice (Campbell et al., 2017). Recent MD simulations of deposition freezing revealed that water first deposits in the form of liquid clusters and then crystallizes isothermally from there (Lupi et al., 2014). So far, there has been no direct experimental evidence of two-step freezing based on probing the molecular structuring of water molecules next to the surface.

In this work, second-harmonic generation (SHG) in total internal reflection (TIR) geometry was used to probe the change in the degree of ordering of water on the surface of mica. SHG is a powerful and simple, compared to sum-frequency generation (SFG), surface–sensitive spectroscopic tool for studying molecules near surfaces and at interfaces (Shen, 1989b; Shen, 1989a). The amplitude and polarization of the generated field, as a function of the polarization of the incident fields, carry information on the abundance and structure of the interfacial molecules between two isotropic media (Jang et al., 2013; Rao et al., 2003; Zhuang et al., 1999). More details on SHG and SFG can be found in the experimental section and in SI. In the system described here, the SHG signal is originated from the nonresonant electric dipolar contribution of the interfacial molecules. The signal response relates to the overall arrangements of the interfacial entities (Fordyce et al., 2001; Goh et al., 1988; Luca et al., 1995) and propotional to the incident field and the second-order nonlinear susceptibility $\chi^{(2)}$ of the interface. When the interface is charged the static electric field due to the charge can induce a third-order nonlinear polarization due to the contribution of the third-order nonlinear susceptibility $\chi^{(3)}$ of the solution (Ong et al., 1992; Zhao et al., 1993). In this work, the contribution of $\chi^{(3)}$ to the total SHG signal has been ignored because there was no significant change in the interface charge with temperature. The change of pH with temperature is very trivial for neutral water (e.g. from pH 7 at 25 °C to pH 7.47 at 0 °C). In addition, this change does not mean that water becomes more alkaline at lower temperatures because in the case of pure water and according to the Le Châtelier's principle there are always the same concentration of hydrogen and hydroxide ions and hence, the water is still neutral (pH = pOH) even if its pH changes. The

pH 7.47 at 0°C is simply the new reference of neutral water pH at 0 °C. In addition, assuming that the surface potential has an influence on the background signal, this will not change even if the pH changes with temperature because the surface potential values of the muscovite basal plane (the surface under study) is pH independent in the range from pH 5.6 to 10 (Zhao et al., 2008).

The results provide new insight into heterogeneous freezing processes and show the suitability of the method for studying current issues relating to ice nucleation, which will contribute to its rapid development in the next years. Initially, I found that the SHG signal drops upon the formation of a thin film regardless of whether the freezing path consists of one− or two−steps and the initially formed phase, liquid or ice, indicating a similar molecular structuring. In addition, I observed a transient SHG signal after immersion freezing. The hygroscopicity of mica is expected to play a role in the described

processes. The hygroscopicity and Langmuir isotherm studies on mica are available in literature but only at room temperature where the sample and environment are at equilibrium (Balmer et al., 2008; Beaglehole et al., 1991; Hu et al., 1995). Such studies at supercooled surfaces are worth to do and could be a topic of future work. An AFM study at 21 °C showed no water absorbed on the surface of mica at RH = 18 % (Hu et al., 1995). The first uniform water phase, of large two-dimensional islands with geometrical shapes in epitaxial relation with the underlaying mica lattice, was observed at RH

= 28 %. The growth of this water phase is completed when the humidity reaches 40 to RH = 50 %. In my experiments on mica it is not possible to detect sub-monolayers, at least at this stage, due to technical reasons mentioned later. In the presented work, only clear steps in the signal were considered.

## 2 Experimental

### 2.1 Materials and setup

All experiments were carried out using MilliQ water (18.2 MΩ·cm). The total organic content in this water is below 4 ppb. Mica samples were obtained from Plano GmbH, Germany. The mica samples were freshly cleaved parallel to the 001 plane in air right before use. The freshly cleaved mica exhibits a wetting surface (on which water was spreading visually). The SHG experiments were conducted using a femtosecond laser system (Solstice, Spectra Physics) with a fundamental beam of 800 nm wavelength, 3.5 mJ pulse energy, ~80 fs pulse width, 1 kHz repetition rate, and a beam diameter of ~2 mm at the

interface. The supercooled SHG setup and the measuring cell are similar to those described in previous publications (Abdelmonem et al., 2015; Abdelmonem et al., 2017). Compared to the setup described in (Abdelmonem et al., 2015), a single fundamental beam incident on the interface was used, Fig. 1, and the SM (S-polarized SHG / 45°-polarized incident) polarization combination was measured. Figure 1 shows the sample and beam geometry. The polarization direction of the incident beam was controlled by a half-wave plate followed by a cube polarizer. The generated signal was collected using a

photomultiplier tube (PMT) placed downstream of an optical system including band pass filters for 400 nm and a polarization analyzer. A sapphire prism was used as an optical coupler to the surface of a thin mica substrate, the basal plane of which was exposed to liquid water or water vapor. The fundamental beam had an incident angle of 15° with the surface normal of the outer side of the prism. Under this geometry, the reflected fundamental (800 nm) and generated SHG (400 nm) beams co-propagate to the sapphire-air interface at the other side of the prism at which both beams are refracted at two

different angles. Only the SHG signal was allowed to reach the detection path. Before starting the measurements, the polarization of the SHG signal generated from water at the surface was analyzed and found to have the expected maxima at S and P polarizations corresponding to an incident 45°-polarized light. The signal was quadratically dependent on the input power. To study mica in TIR geometry, an index matching gel (IMG) from Thorlabs (G608N3, RI~1.45186 at 800nm) was used to fix the mica sample on the hypotenuse of the sapphire prism. The freezing point was not specified by the

manufacturer but tested in the lab. At least the gel was not frozen until -45 °C. A detailed description of sample geometry and the selection of the IMG was published in (Abdelmonem et al., 2015). Less than 15% of the laser output power was

coupled to the setup to not destroy the IMG. The fluctuation in the signal due to reducing the laser power limited the sensitivity of the system and it was not possible to monitor minor changes which may arise from pre-adsorption of sub-monolayers at temperatures higher than the dew point or freezing point.

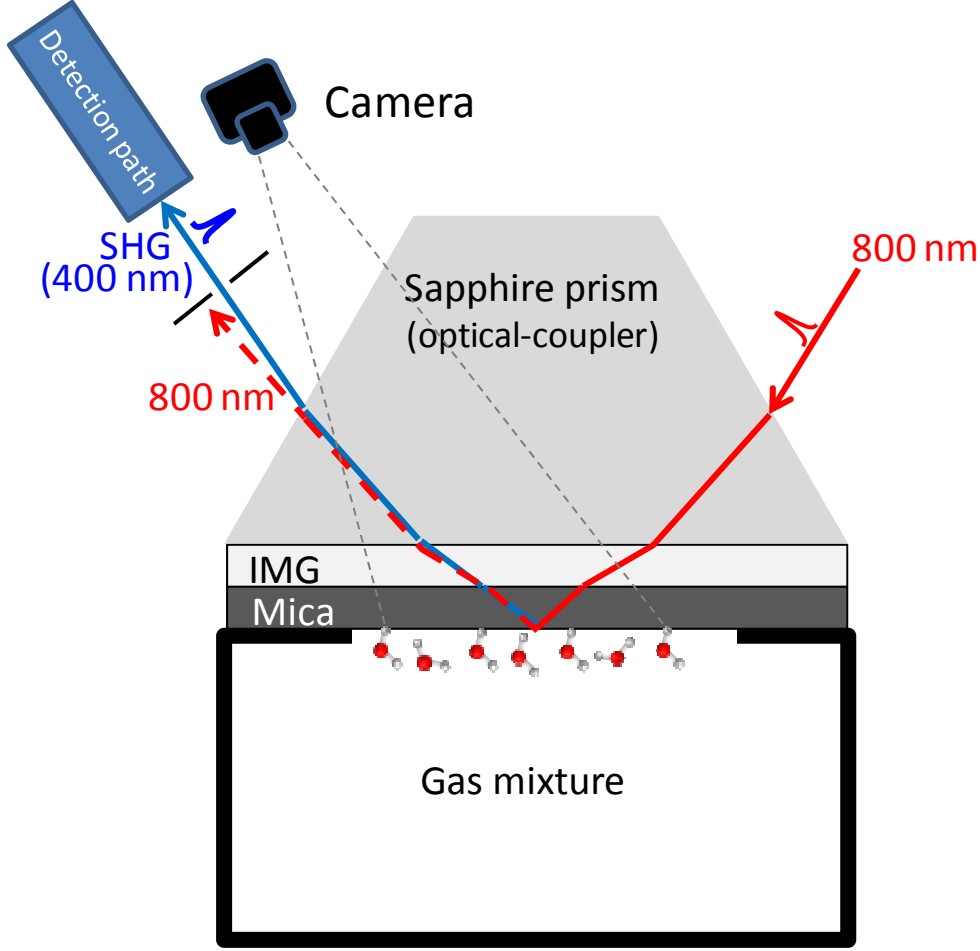

**Figure 1: The sample and beams geometry. See text for details**

A homemade temperature-controlled environmental chamber, Fig. 2, was integrated into the setup. A commercially available cold-stage (Linkam model HFS-X350) was used after modifying the housing to accommodate the SHG setup. The sapphire prism was place in a cupper adaptor which was fixed on the silver block of the Linkam cold-stage. The substrate of interest was sealed to a circular opening of 8 mm in diameter in a teflon cell which was being purged with the sample gas during the experiments. The cold-stage can perform controlled heating and cooling ramps, applied to the silver block, at rates between 0.01 and 100 °C/min. Temperature stability of the cold-stage is better than 0.1 K. The temperatures of the air inside the cell and the sapphire prism top and bottom were measured using four-wire-Pt100 elements. The temperature of the probed spot on the surface was considered to be the average of the sample top and sample bottom temperatures. However, it should be emphasized that the exact onset condition of freezing is not the focus of this work, but rather the study of the qualitative behavior of water molecules during freezing on the different paths. During the experiments, the gas box was filled with $N_2$ gas to avoid condensation on the outer surfaces of the prism during cooling. The humid air pumped to the measuring cell was obtained by mixing dry gas and 100 % humid gas with different ratios at 21 °C using two mass flow controllers (Tylan 2900). The continuous flow of the gas (either dry or humid) during the experiment set the temperature inside the cell to 21 °C ± 0.5 °C. The corresponding fluctuation of the relative humidity was less than 0.2%. The corresponding fluctuation in the dew point, at RH = 5 ±0.2 for instance, was ±0.5 °C. The gas mixing ratio versus RH was calibrated by setting a mixing ratio, cooling the sample and recording the condensation/freezing temperature at which the reflectivity at an angle equals to

the critical angle of TIR for air-mica interface starts to drop due to the violation of TIR condition. This temperature was used to define the corresponding RH using Arden Buck equation.

The same method was used to differentiate between liquid-film, liquid-bulk, transient ice, and stable ice in this work. The border between a film at the solid-air interface and a bulk at the solid-water (or -ice) was defined experimentally by the point where the intensity of a TIR reflected light from the solid-air interface drops due to the violation of TIR condition when the refractive index of the contact medium changes drastically from that of air ($n_a$=1) to one of those of water or ice ($n_{w \text{ or } i}$ > 1.3). Whether the contact medium is liquid or ice, this was defined by the change in the light scattering, observed at a CCD camera (Guppy F-036 Allied Vision Technology with LINOS Macro-CCD Lens 0.14x (1:7) f4) placed close to the detection path, Fig. 1. After immersion freezing, there was a rapid increase in the signal and then slow decrease. The maximum after the rapid increase was defined as the "transient ice" data point. After reaching a maximum, the signal was decreased with time until stabilized after certain time. This stabilized signal was defined as the "stable ice" data point.

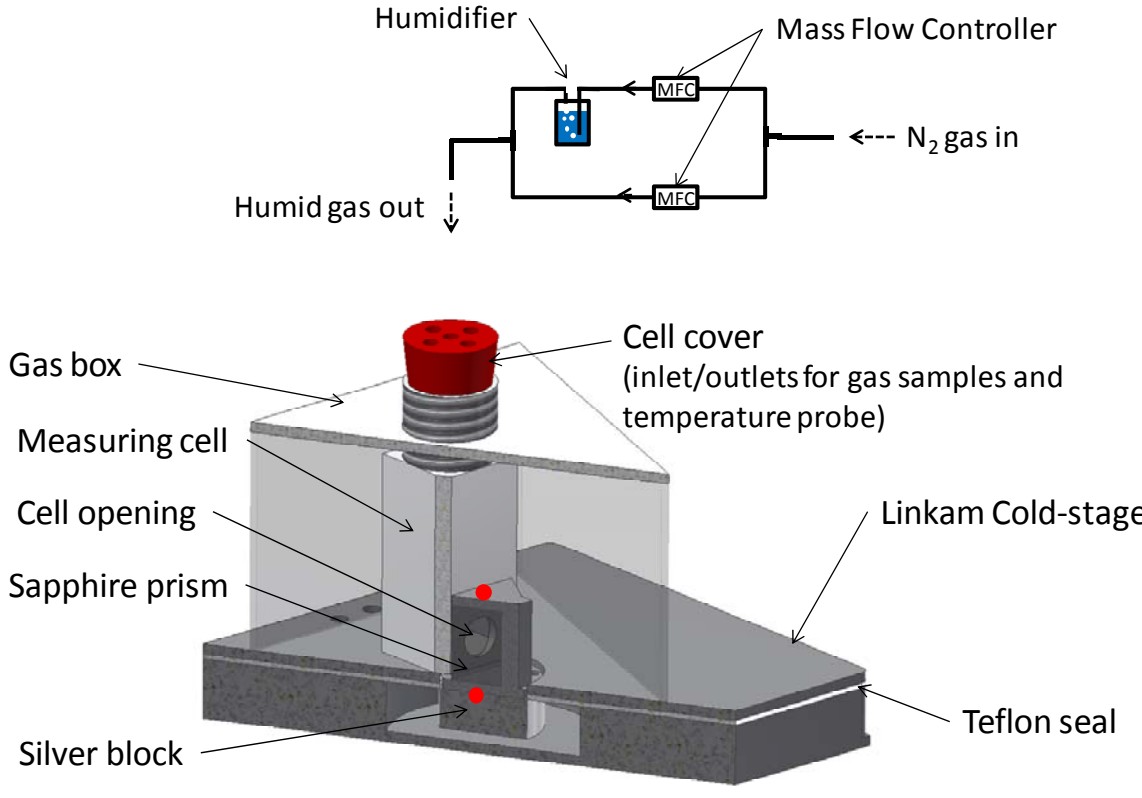

**Figure 2: The temperature-controlled environmental chamber configuration, humidification and temperature sensors. The red dots are the position of the sample temperature sensors. There is a temperature sensor inside the measuring cell (not shown)**

A control software was developed to adhere to a predefined temperature profile and to measure the SHG signal and the temperature of the substrate. Temperature profiles were repeated several times for each sample to test reproducibility. In each run, the sample was heated to 110 °C while purging with $N_2$ gas (99.9999 %) to evaporate any residual water, then cooled down to 0 °C at a rate of 10 °C/min and then down to the heterogeneous freezing point at a rate of 1 °C/min. This cooling profile was the same for all runs to allow for logical comparison. An experiment with a mica-$N_2$ gas interface was carried out to ensure that the change in the refractive indices of the sapphire prism, IMG, and mica substrate with temperature have no significant effect on the resulting SHG signal in the range of freezing temperatures observed in this work, Figure S4 in SI.

As mentioned above, the incident angle from air of the fundamental beam was adjusted to 15° with respect to surface normal of the outer side of the sapphire prism. The corresponding incident angle on the mica-air or -water interface was ~ 63.4°, which is higher than the critical angles of TIR for mica-air (~39.7°) and mica-water (~59.2°) interfaces. This guaranteed a TIR condition regardless of any changes in the Fresnel factors caused by the change of refractive indices with changing temperature. The advantage of using SM polarization combination is its dependence on only one non-vanishing nonlinear susceptibility tensor element ($\chi_{yyz}$), (Shen, 1989a; Zhuang et al., 1999), at any working angle which makes it a direct probe of the degree of order of the molecules at the interface.

The SHG signal is mainly produced by all polarizable species within the SHG-active region as long as the inversion symmetry is broken. The polarizable species at the surface of interest are the water molecules and the surface OHs. The contribution of water molecules are limited by the penetration depth inside the second medium (air, liquid water or ice). Under the optical geometry described above, the calculated penetration depths are about 130 nm, 328 nm, and 253 nm for air, liquid water bulk, and ice bulk as contact media respectively. Under the thermodynamical conditions of the presented work, the thickness of the ice (or water) layer should exceed 1 μm within 1 sec after nucleation which is far beyond the penetration depth of the evanescent field. Therefore, although the exact thickness cannot be determined in this setup it does not affect the probed signal, by the Two-Interface Problem, because the second interface (ice-air or water-air) is not within the focal volume of the pump beam and the non-resonant signal is coming exclusively from the first interface (water-solid or ice-solid). For the ice layer thickness, the reader is referred to the calculations of the growth velocity of a solidification front normal to the ice surface provided in the Supplementary Materials of the recent work of Kiselev et al. (Kiselev et al., 2016). These calculations were taken from numerous works of Libbrecht (Libbrecht, 2003; Libbrecht, 2005). For calculations of growth due to condensation, the reader is referred to the Aerosol Calculator Program (Excel) by Paul Baron which is based on equations from (Willeke and Baron, 1993; Hinds, 1999; Baron and Willeke, 2001). The signal was not collected until it became stabilized and therefore, it was assumed that the ice layer after deposition was uniform at the surface covered by the laser spot.

## 3 Results and discussion

The relative humidity (RH) of the purged gas was set to specific values in different runs to allow for different freezing modes on the surface of mica. Figure 3a shows the change in normalized Fresnel factor-corrected (see SI for details) SHG intensities under SM polarization combination for deposition freezing (DF) at -38 °C (black solid line and circles), -21 °C (red solid line and empty squares), and at -12.5 °C (green solid line and empty triangles) labeled by DF1, DF2, and DF3, respectively. In DF1, the cell was filled with $N_2$ gas and the sample was cooled down to a temperature, -38 °C, far below the dew point at RH = 5 % (-21 °C). At -38 °C, the cell was purged with humid air of RH ~ 5 %. An ice−film formed immediately on the surface, reflected by a drop in the SHG signal compared to that of air-mica interface. In DF2 and DF3, the cell was purged continuously with humid air of RH = 5 and 11 %, respectively, and then cooled down until freezing and growth of ice were observed on the surface. Deposition freezing and the formation of an ice−film started at -21 and -12.5 °C for RH = 5 % (DF2) and 11 % (DF3), respectively. The results show a drop in the SHG signal with respect to the signal of the mica-air interface upon freezing for the three cases DF1, DF2, and DF3. However, the relative signal drop for DF1 differs from those of DF2 and DF3. DF2 and DF3 were observed at temperatures equal to the dew points at the preset RHs, indicating two-step nucleation, first condensation, and then freezing. The coincidence of the SHG signals of the thin ice−film formed in DF2 and DF3 indicates similar degree of order of water on the surface in two−step deposition freezing regardless of the onset temperature. This means that at temperatures above -38 °C, growth of ice on mica apparently is a two-step process: Water first condenses and then freezes. This confirms on the molecular level the two-stage nucleation hypothesis

which suggests that a nucleus forms as a liquid cluster and then freezes (Layton and Harris, 1963; Lupi et al., 2014). Further pumping of the gas mixture to the measuring cell allows for the growth of the ice–film by diffusion. The resulting ice–bulk shows a further drop in the signal for DF2 and DF3. This drop was not observed for DF1, thus indicating a major difference in the spectroscopic behavior of ice between one–step and two–step deposition freezing. To ensure that the lack of change in

the SHG signal after the growth of an ice–film to ice–bulk in one-stage nucleation, DF1 figure 3a, is not an artefact, DF1 and DF2 were compared using a different system (sapphire–water interface, Fig. 3b). Figure 3b shows the change in SHG intensity at the surface of sapphire for DF at -38 (black solid line and circles) and at -23 °C (red solid line and empty squares) labeled by DF1 and DF2, respectively. As in the case of mica, the drop in SHG intensity after the formation of an ice–film was followed by another drop upon further pumping of humid air of RH = 5 % in the two-step freezing process

(DF2), but not in one-step (DF1) freezing.

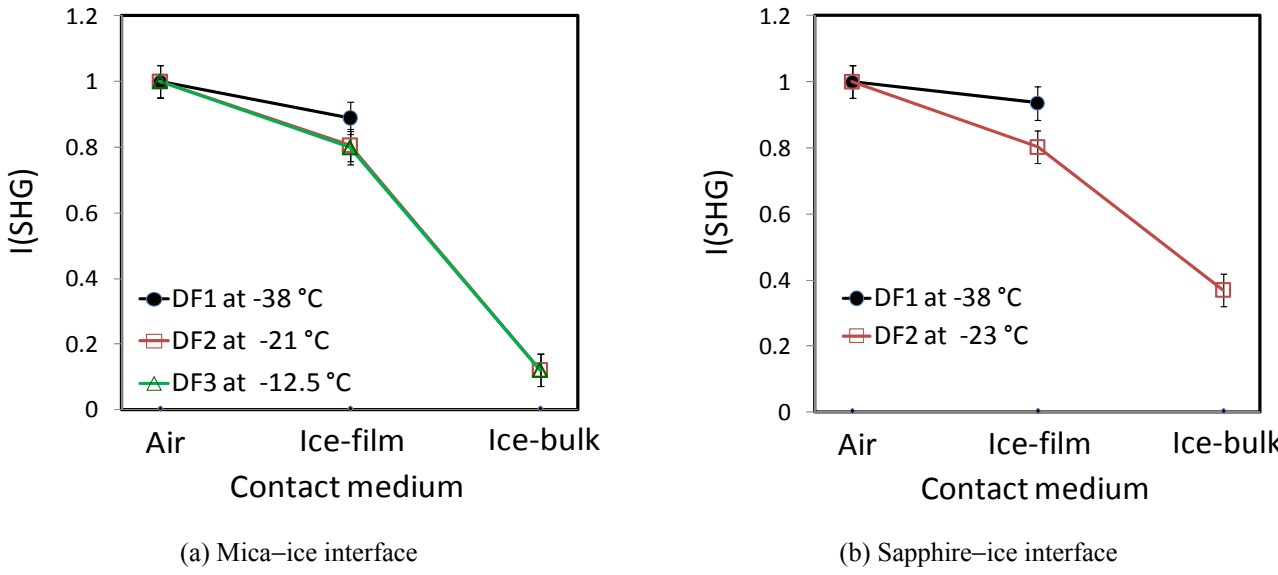

(a) Mica–ice interface                                     (b) Sapphire–ice interface

**Figure 3: SHG intensity measured at the surfaces of mica (a) and sapphire (b) in contact with air, ice–film, and ice–bulk, respectively, collected in SM polarization combination during the three different cooling cycles, DF1: Sample cooled down first to - 38 °C under dry $N_2$ gas , followed by pumping water vapor of RH = 5 % to the measuring cell. DF2: Sample cooled down under flow of water vapor of RH = 5 %. DF3: Sample cooled down under flow of water vapor of RH = 10 %. The signal is Fresnel-**
**corrected and normalized to the air value. All connection lines between points are just for guiding the eyes.**

Figure 4 shows three different freezing experiments at different RHs. The gas RH was adjusted to allow liquid condensation (LC) during cooling at temperatures higher than those of deposition freezing. Constant pumping of humid air at RH = 20, 30, and 40 % and cooling down resulted in the formation of stable liquid films at -3.5, 3, and 7 °C, respectively. At all RH values, the SHG signal drops down upon the formation of a liquid–film by LC. The relative drop of the signal with respect to

the air signal is similar to that observed in the DF experiments, Figure 3a. Comparing Figures 3a and 4, the SHG signals are in the same range regardless of the film phase (liquid or ice). By further pumping of humid air after LC, liquid–bulk forms at the surface with a signal that is lower than that of the liquid–film. This is mostly due to the contributions from the few secondary layers of the interfacial water. Further cooling of the sample in contact with liquid–bulk causes water to freeze by immersion freezing (IF). The observed IF temperatures for IF1, IF2, and IF3 are similar and center around -11 °C ± 1 °C

which is within the range of IF temperatures observed for freezing of bulk water in contact with the surface of mica in former studies (Abdelmonem et al., 2015; Anim-Danso et al., 2016). IF produced a transient ice phase with SHG intensity higher than that of the interfacial water of the bulk liquid. The lifetime of the transient phase is around one minute, Figure 5. The values of the SHG intensities plotted in Figure 4 for transient ice–bulk are the peak values found on the transient curves shown in Figure 5 after Fresnel factor corrections and normalization to the mica-air signal. The transient phase may have had

peak values higher than those obtained from Figure 5, but they were not detected due to the fast signal decay right after

nucleation and the limited time resolution of signal detection of about 2.5 sec. A transient phase lasting for several minutes was reported very recently by Lovering et al. using SFG at a water-silica interface (Lovering et al., 2017). They suggested a transient existence of stacking-disordered (non-centrosymmetric) ice during the freezing process at water-mineral interfaces. Anim-Danso et al. also observed such transient ice lasting for a few tens of seconds in SFG experiments at a high-pH (9.8)

5    solution-sapphire interface. They suggested that charge transfer and the stitching bilayer are perturbed at high pH, which leads to a decrease in SFG intensity. The present work shows that the transient ice occurs at neutral pH on mica surface and has a significant nonresonant component which is observable with the simple SHG technique. Apparently, the lifetime of the transient phase depends on the substrate and probably on the liquid–bulk size and might play a big role in the ice nucleation ability of the surface. However, this requires a comparative study involving different substrates, which will be subject of

10   future work.

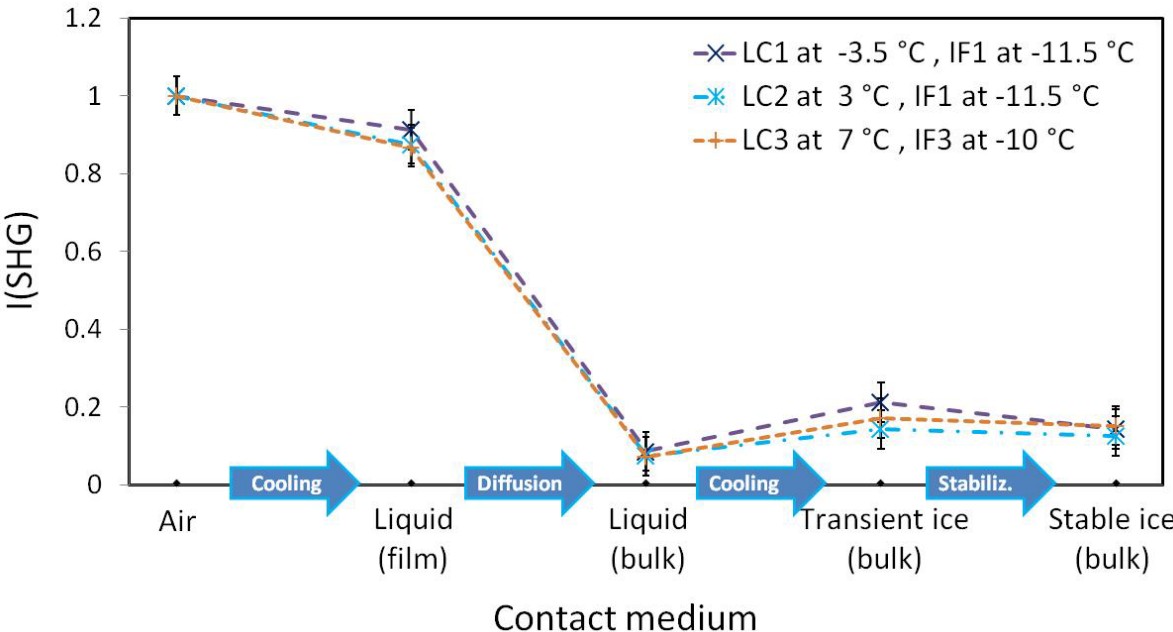

**Figure 4: SHG intensity measured at the surface of mica in contact with air, liquid–film, liquid–bulk, transient ice–bulk, and stable ice–bulk, respectively, collected in SM polarization combination during three different cooling cycles, LC1 IF1: Sample**
15   **cooled down under flow of water vapor of RH = 20 %. LC2 IF2: Sample cooled down under flow of water vapor of RH = 30 %. LC3 IF3: Sample cooled down under flow of water vapor of RH = 40 % (see text for details). The signal is Fresnel-corrected and normalized to the air value. All connection lines between points are just for guiding the eyes.**

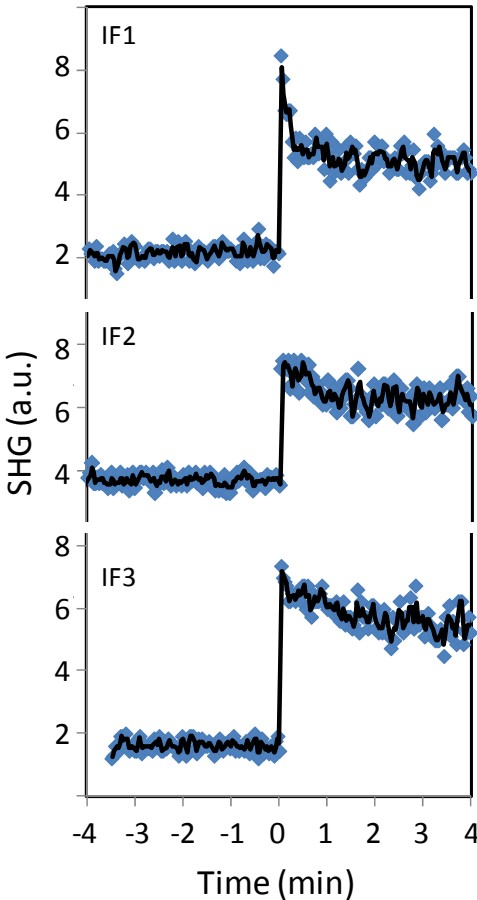

**Figure 5: Typical variation in the SHG intensity around the immersion freezing points for IF1, IF2, and IF3 during the cooling experiments**

Finally, I would like to comment on the drop, rather than increase, of the signal upon adsorption of water (or ice) on the
surface (of either mica or sapphire). Since SHG response reflects the overall arrangements of the polar entities at the
interface between two isotropic media (Fordyce et al., 2001; Goh et al., 1988; Luca et al., 1995), signal intensity is expected
to increase when a single (or few non-centrosymmetric) layer(s) of water or ice is(are) formed at the surface. It is clear from
Figures 3 and 4, however, that SHG intensity decreases upon deposition freezing, condensation, and growth of liquid layers
by diffusion. This can be explained by phase interference between two signals originating from two different interfacial
groups of opposite dipole moments: surface-OH points out of the surface and water points to the surface. A cleaved mica
surface exhibits a disordered hexagonal arrangement of Si- (partly Al-) doubly coordinated O atoms in the outermost layer
(Ostendorf et al., 2008). As an aluminosilicate mineral, this surface may protonate immediately in contact with the ambient
air (forming silanol and aluminol groups at the surface). In this case, the mica-dry air signal can mainly originate from the
surface hydroxyl groups (dangling−OH) which are naturally pointing out from the surface. Since mica surface is inherently
negatively charged, the interfacial water will point to the surface and thus have a phase opposite to that of the surface−OH
groups, as was reported by Shen and co-workers (Zhang et al., 2008) on negatively charged surfaces. This well explains the
ostensible decrease in the overall signal upon deposition and condensation. When the surface of mica is covered with bulk
water of a pH ~ 7, it becomes totally deprotonated (no free−OH). One in four Si in the tetrahedral layers of mica is randomly
substituted for Al (Christenson and Thomson, 2016). The result is that the majority of the surface hydroxyl groups of a
cleaved basal plane are silanol and therefore the surface deprotonation is significantly determined by silica. The point-of-
zero-charge (pzc) of silica lies between its respective isoelectric points pH 2 − 3 (Hartley et al., 1997; Iler, 1979; Scales et
al., 1992) and hence the silanols in silica (Si-OH) deprotonate totally in the presence of neutral water (pH ~ 7). Under this
condition the SHG signal comes exclusively from the interfacial water molecules between the surface and liquid-bulk. This
defines a new reference of the signal generated at the new interface (liquid-solid rather than gas-solid). The increase of the

signal afterwards indicates more structuring of either the interfacial water before freezing, like shown in (Abdelmonem et al., 2015) or the interfacial ice after freezing like shown here. It is worth to mention that, the presence of free surface hydroxyl groups on cleaved mica is poorly discussed in literature. Miranda et al. have indirectly referred to their (hydroxyl groups) existence in their work on mica-water vapor interface using SFG (Miranda et al., 1998). They used deuterated water ($D_2O$) in the SFG experiments to avoid confusion of the hydroxyl stretch modes in the spectrum from both water and mica. Maslova et al. have discussed it a bit explicit in their work on surface properties of cleaved mica and they assumed that hydroxyl groups of the basal plane are not reactive (Maslova et al., 2004).

**Conclusion**

In summary, I used a simple SHG setup to discriminate and describe three different freezing paths on the surface of mica. The results decide on, and confirm on the molecular level, the previous speculations about the existence of two-stage deposition ice nucleation at temperatures above -38 °C. One-step and two-step deposition freezing of a thin film of ice show water structure similar to that of a thin film of liquid water. When a liquid–bulk freezes at the surface by immersion freezing, there is a transient non-centrosymmetric ice phase of high non-resonant SHG signal. The lifetime of the transient phase is suggested to be substrate–dependent and expected to affect ice nucleation efficiency. The presented results open up new horizons for the role of aerosol surfaces in promoting and stabilizing heterogeneous ice nucleation. They provide novel molecular–level insight into different ice nucleation regimes using a simple spectroscopic technique. Investigating the structuring of water molecules upon freezing next to solid surfaces is crucial to many scientific areas, such as atmospheric physics and chemistry, hydrology, and environmental and industrial applications.

The work demonstrates the worthwhile of investigating different ice nucleation processes and water structuring upon freezing on the molecular-level using SHG spectroscopy. The manuscript is considered as a cornerstone of future complementary studies involving other surfaces and other techniques to precisely investigate the layer thickness, the surface morphology effect, the cooling rates ...etc. The difficulty of characterizing monolayer films rose from the use of the IMG which required reducing the power. An alternative would be to approach the surface from the air side which then has the disadvantage of a week SHG signal and Two-Interface Problem. These are challenges which will be tackled in future works.

**Acknowledgements**

The work is funded by the German Research Foundation (DFG, AB 604/1-1). The SHG setup was funded by the competence area "Earth and Environment" of KIT (start-up budget 2012). The author is grateful to Dr. A. Kiselev for his scientific discussions on ice growth and to Prof. Dr. T. Leisner, Dr. J. Lützenkirchen, Dr. C. Linke and Mrs. M. Schröder from the KIT for their support.

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
