# Peer review of "Direct molecular level characterization of different heterogeneous freezing modes on mica – Part 1"

_Atmospheric Chemistry and Physics, 2017_

## Referee Comment (RC1) · Anonymous Referee #1 · 11 May 2017

This manuscript describes SHG measurements during water-mica interactions recorded at low temperature. The sparseness of the SHG data, the incorrect interpretation thereof, and the lack of complementary measurements lead this reviewer, unfortunately, to recommend rejection of the work. 1) The document is in a poor state of editing, with many grammatical errors that substantially distract from evaluating it. 2) The document contains false statements re: the origin of the SHG response. The system is probed off resonance, which means that all terms contributing to the response are purely real. Statements like "the SHG signal is originated from the the nonresonant OH stretching vibrations at the interface" are simply incorrect and reflect a fundamental misunderstanding of the signal generation process by the author. The

signal is produced by all polarizable species within the SHG-active region. Unfortunately, the SHG active region is neither characterized nor defined in this work, making the signal interpretation at best appear as creative writing. 3) The interference that is briefly alluded to in the final paragraph is not quantified, even though the changes in the SHG responses shown in the three figures are produced by said interference, in addition to changes in surface potential that occur during the experiments. The author is encouraged to read and understand the recent work on nonlinear optical interference in thin-layer systems by Massari (J. Phys. Chem. Lett., 2016, 7 (1), pp 62–68) and on the chi(2) and chi(3) phase interference by Wang, Geiger, and Eisenthal (Nature Communications, 7, 13587, 2016). 4) The work requires additional information on ice layer thickness, on the uniformity of the ice layers across the 2 mm laser spot, and it requires verification wether the SHG signal depends quadratically on input power. The polarization states of the SHG responses during the various stages of the experiments should also be determined. 5) Connections of any results and/or discussion presented to the scientific motivation provided are not made except for two generic statements ("They provide novel molecular−level insight into different ice nucleation regimes..." and "Investigating the structuring of water molecules upon freezing next to solid surfaces is crucial to many scientific area...") which are broad and sweeping. In sum, this work is far too preliminary to be reconsidered. As such, this reviewer recommends rejection, with the hope that the author will write a new document that addresses the points made above in a new submission elsewhere.

---

## Referee Comment (RC2) · Anonymous Referee #2 · 22 May 2017

This manuscript by Abdelmonem studies the deposition freezing of water on mica surfaces using SHG. The author claimed that two sub-deposition nucleation modes were identified but this claim was poorly supported by the experimental data. The manuscript also contains some incorrect statements about SHG. This reviewer does not recommend its publication in ACP.

(1) The author stated that "the SHG signal is originated from the nonresonant OH stretching vibrations at the interface". (Line 8 on page 4) This statement does not make sense. A nonresonant signal is by definition not to be associated with a particular vibrational mode. (2) Mica is birefringent. As the laser beam goes through the mica, its polarization may not be linear any more. "The advantage of using SM polarization

combination is its dependence on only one non-vanishing nonlinear susceptibility tensor element (line 33 on page 3)" is likely not what has happened in the experiment. (3) The index-matching gel of unknown chemical composition is a concern. While the gel helps to obtain the TIR condition, the gel/mica and gel/sapphire interfaces may produce SHG signal. The author may want to look into the freezing temperature of the gel too. (4) Figure 1 and 2 should be real-time plots similar to Figure 3. (5) Terms such as "liquid (film)", "liquid (bulk)", "transient ice", "stable ice" used in Figure 2 should be experimentally defined. (6) The cooling rate dependence should be investigated. (7) One fundamental issue of SHG is that SHG intensity can be difficult to interpret. For example, why is there a ~80% SHG drop from the "air" to the "bulk liquid" in Figure 2? What does the SHG measure? (8) Some conclusions made in the manuscript are not well supported by the SHG data. For example, line 8 on page 4: "The coincidence of the SHG signals of the thin ice−film formed in DF2 and DF3 indicates identical structuring of water on the surface in two−step deposition freezing regardless of the onset temperature." SHG simply cannot provide the structural information of water. The same SHG intensity does not necessarily mean the same water structure.

---

## Short Comment (SC1) · 24 May 2017

I have read with interest the recent contribution of A. Abdelmonem to Atmospheric Chemistry and Physics Discussions. Second-harmonic generation (SHG) spectroscopy has the potential to be a powerful tool for investigating freezing at interfaces. However, there are a number of points regarding the mica surface and previous work with mica that should be taken into consideration in any revised manuscript. In particular, these points may lead the author to reconsider some of his interpretation of the results.

1. The two-step nucleation process postulated by Campbell et al. in 2013 involved various organic liquids crystallising from vapour on mica surfaces, but only in surface

features such as cleavage steps, cracks and pockets. A later study (Campbell et al., 2017)) has confirmed the two-step process for the organic liquids, and strongly suggested a two-step process for water and ice, although conclusive proof could not be obtained for water and ice.

2. The material used by Layton and Harris in their 1963 paper (the reference is incomplete) was not muscovite mica but synthetic fluorophlogopite, which is similar in structure but has the hydroxyl groups replaced by fluorine. Moreover, neither is a metal oxide as stated in the abstract, but they are both layered aluminosilicates.

3. The basal (cleavage) plane of mica consists of distorted hexagons of oxygen atoms, and there are no Si or Al atoms in the outermost layer. Moreover, the hydroxyl groups are below the oxygen hexagons, so there are none available for hydrogen bonding to adsorbed water molecules. This is explained in a recent review (Christenson and Thomson, 2016) which gives numerous references to the original literature on the structure determination of muscovite mica by X-ray diffraction.

4. The simulation by Odelius et al. (1997) suggested that the mica surface is covered by a network of hydrogen bonds between water molecules only, with no free water hydroxyls. However, these results have been called into doubt by more recent simulations (Wang et al., 2005; Malani and Ayappa, 2009), and in a density-functional study that found no evidence of 2D-ice on mica, but that the properties of water on the surface are dominated by hydration of potassium ions (Feibelman 2013).

5. The study of air-cleaved mica and water vapour (as is the case here for the measurements in water vapour) is complicated by the fact that the surface potassium ions are free to diffuse along the surface (as potassium carbonate), which may even result in the formation of crystallites on the surface in dry conditions. This was first shown by Christenson and Israelachvili in 1987, and was discussed in Balmer et al., 2008. The implications for studying mica in humid atmospheres was summarised in the 2016 review mentioned under point 3. Mica immersed in bulk water does not suffer from

these problems, of course, as any potassium carbonate dissolves and the nature of the surface ions is determined by the pH and any residual electrolyte in the solution. The mobility of the potassium ions (as potassium carbonate) does not necessarily alter dramatically any measurement of average surface properties, as is the case with SHG, and at high humidities the potassium will be widely dispersed across the surface. However, the surface mobility of the potassium will necessarily decrease at lower temperatures, and there may be a dependence on the history of the mica surfaces. To summarise, it should be borne in mind that what is adsorbed to the mica surface is a thin film of aqueous potassium carbonate, the concentration of which varies with humidity and temperature, rather than pure water.

New references Christenson, H. K. and Thomson, N. H.; The Nature of the Air-cleaved Mica Surface, Surf. Sci. Rep. 71, 367-390, 2016. Christenson, H. K. and Israelachvili, J. N.: Growth of Ionic Crystallites on Exposed Surfaces, J. Colloid Interface Sci. 117, 576-577, 1987. Campbell, J. M., Meldrum, F. C. and Christenson, H. K.; Observing the Formation of Ice and Organic Crystals in Active Sites, Proc. Natl. Acad. Sci. USA, 114, 810-815, 2017. Feibelman, P. J.; K+-hydration in a low-energy two-dimensional wetting layer on the basal surface of muscovite, J. Chem. Phys. 139, 074705, 2013. Wang, J., Kalinichev, A. G.,Kirkpatrick, R. J. and Cygan, R. T.; Structure, Energetics, and Dynamics of Water Adsorbed on the Muscovite (001) Surface: a Molecular Dynamics Simulation, J. Phys. Chem. B 109 2005, 15893-15905, 2005. Malani,A. and Ayappa, K. G.; Adsorption Isotherms of Water on Mica: Redistribution and Film Growth, J. Phys. Chem. B 113 1058-1067, 2009.

H. K. Christenson School of Physics and Astronomy University of Leeds Leeds LS2 9JT U. K.

---

## Author Comment (AC2) · 13 Jun 2017

RC: 1) The document is in a poor state of editing, with many grammatical errors that substantially distract from evaluating it.

*AC: As the author is not a native English speaker the manuscript was revised by the "language service department" at the KIT before submission. Anyways, if the journal requires professional lingual editing by the journal itself or somewhere else, the author will certainly consider that.*

RC: 2) The document contains false statements re: the origin of the SHG response. The system is probed off resonance, which means that all terms contributing to the response are purely real. Statements like "the SHG signal is originated from the the nonresonant OH stretching vibrations at the interface" are simply incorrect and reflect a fundamental misunderstanding of the signal generation process by the author. The signal is produced by all polarizable species within the SHG-active region. Unfortunately, the SHG active region is neither characterized nor defined in this work, making the signal interpretation at best appear as creative writing.

*AC:*

*Indeed the statement "the SHG signal is originated from the nonresonant OH stretching vibrations at the interface", in page 2 line 19, is incorrect. However, it should be clear that this is not a "fundamental misunderstanding of the signal generation process by the author" but rather an oversight, wording failure, mixed with the definition of water stretching signal in SFG. This should become clear to the reader when reaching to page 6 line 9 when the author started discussing interpretations in terms of "electric dipolar contribution" as an origin of the signal. (This oversight will be corrected in the revised version)*

*The polarizable species at the surface are the water molecules and the surface OHs. The experiments were carried out in total internal reflection geometry with an incident angle of 15° from air to the prism side\*. The SHG-active region is limited by the penetration depth of the evanescent wave in the second medium (air, liquid water or ice). The calculated penetration depths are about 130 nm, 328 nm, and 253 nm for air, liquid water bulk, and ice bulk as contact media respectively.*

*\* Note: there was a typo in the corresponding incidence angles at interfaces in the original manuscript. This will be corrected in the revised version.*

RC: 3) The interference that is briefly alluded to in the final paragraph is not quantified, even though the changes in the SHG responses shown in the three figures are produced by said interference, in addition to changes in surface potential that occur during the experiments. The author is encouraged to read and understand the recent work on nonlinear optical interference in thin-layer systems by Massari (J. Phys. Chem. Lett., 2016, 7 (1), pp 62–68) and on the chi(2) and chi(3) phase interference by Wang, Geiger, and Eisenthal (Nature Communications, 7, 13587, 2016).

*AC: The referee speaks here about the "Two-Interface Problem" and refers to the work of Massari (J. Phys. Chem. Lett., 2016, 7 (1), pp 62–68). Certainly, the author ignored this effect, and the mentioned paper is not relevant to the presented work, for the following reasons:*

1. *The mentioned work describes the interference between two resonant signals generated at the two interfaces (on two sides) of a well controlled thin-film prepared for an organic thin-film field-effect transistor. The two interfaces are well within the focal volume of the pump beams.*

2. *The signal interference occurs between two oppositely oriented groups, of the same type, at both sides of the thin-film*

3. *In contrast, ice (or liquid) -film growth on a surface is completely different. The growth depends mostly on the atmosphere saturation for a given temperature and pressure, and the thermal conductivity of the substrate material. When the air is supersaturated, the growth rate is very fast after nucleation and is very difficult to control.*

4. *In the presented work the second interface (ice-air or water-air) is not within the focal volume of the pump beam (please remember that the signal in the presented work was collected in TIR geometry) and the non-resonant signal is coming exclusively from the first interface (water-solid or ice-solid) due to the following: Under the thermodynamical conditions of the presented work, the ice (or water) layer thickness exceeds 1 μm within 1 sec which is far beyond the penetration depth of the evanescent field (see values above). The author would like to draw the attention of the referee to a related details on the growth velocity of a solidification front normal to the ice surface in the Supplementary Materials of the work of one of our KIT groups (Kiselev et al., 2016). These calculations were taken from numerous works of Libbrecht (Libbrecht, 2003; Libbrecht, 2005). These articles will be cited in the revised version along with the corresponding discussion. For calculations of growth due to condensation, the reader is referred to the Aerosol Calculator Program (Excel) by Paul Baron which is based on equations from (Willeke and Baron, 1993; Hinds, 1999; Baron and KlausWilleke, 2001). These will also be cited in the revised version with the corresponding discussion.*

*The referee has also referred to the great paper on Phase-referenced nonlinear spectroscopy of the $\alpha$-quartz/water interface by Wang, Geiger, and Eisenthal (Nature Communications, 7, 13587, 2016) which shows that the absorptive (imaginary) and dispersive (real) terms of chi(2) chi(3) may mix. Again the author sees no significant relevance[†] to the presented work for the following reasons:*

1. *Regardless of the interesting role of the anisotropy of $\alpha$-quartz to generate phase-referenced SHG signal, the mentioned paper discusses two extreme pHs (pH 3 and pH 11.5 which have 8.5 orders of magnitude hydrogen ion activity factor) and different ionic strengths. In contrast, in the presented work, neither the pH nor the ionic strength was changed. The author used deionized water with pH ~ 7.*

2. *The change of pH with temperature is very trivial for neutral water (e.g. from pH 7 at 25 °C to pH 7.47 at 0 °C). In addition, this does not mean that water becomes more alkaline at lower temperatures because in the case of pure water and according to the Le Châtelier's principle there are always the same concentration of hydrogen and hydroxide ions and hence, the water is still neutral (pH = pOH) even if its pH changes. The pH 7.47 at 0°C is simply the new reference of neutral water pH at 0 °C.*

3. *Assuming that the surface potential has an influence on the background signal, this will not change even if the pH changes with temperature. It was found that the surface potential*

*values of the muscovite basal plane (the surface under study) is pH independent in the range from pH 5.6 to 10 (Zhao et al., 2008).*

*† Probably the author should have mentioned the chi(3) mechanism for interfacial potential-induced SHG which is an important factor in such experiments and has been originally established by the group of Eisenthal (Zhao et al., 1993; Ong et al., 1992). This will be considered in the revised manuscript.*

RC: 4) The work requires additional information on ice layer thickness, on the uniformity of the ice layers across the 2 mm laser spot, and it requires verification wether the SHG signal depends quadratically on input power. The polarization states of the SHG responses during the various stages of the experiments should also be determined.

*AC: As mentioned above, the ice layer thickness exceeds the penetration depth of the evanescent field at the time when the signal was collected. The exact thickness cannot be determined in this setup, however it does not change the probed signal. (Will be mentioned in the revised manuscript)*

*At the deposition, and also condensation, events the signal was not collected until it became stabilized and therefore, it was assumed that the ice layer is uniform at the surface covered by the laser spot. (Will be mentioned in the revised manuscript)*

*It was verified that $I(400) \alpha I(800)^2$. (Will be mentioned in the revised manuscript)*

*As mentioned in page 3 line 26m, the SHG was collected S-polarized.*

RC: 5) Connections of any results and/or discussion presented to the scientific motivation provided are not made except for two generic statements ("They provide novel molecular-level insight into different ice nucleation regimes..." and "Investigating the structuring of water molecules upon freezing next to solid surfaces is crucial to many scientific area...") which are broad and sweeping. In sum, this work is far too preliminary to be reconsidered. As such, this reviewer recommends rejection, with the hope that the author will write a new document that addresses the points made above in a new submission elsewhere.

*AC: Certainly, the aim of the work was to demonstrate the worthwhile of investigating different ice nucleation processes and water structuring upon freezing on the molecular-level using SHG spectroscopy. This has been successfully demonstrated although conclusive description of ice nucleation on mica has not been obtained which is not expected at this stage. The manuscript is more like a "letter" or "short communication". However, ACP does not provide an individual category for particularly short papers because the process of peer review and publication in ACP is inherently efficient and rapid for all types of manuscripts without artificial length restrictions. A complementary study involving other techniques and sophisticated studies will follow this paper which is considered as a cornerstone for future studies. Probably a better title of the manuscript could be "Direct molecular level characterization of different heterogeneous freezing modes on mica – Part 1"*

**NOTE: The modified text will be posted in a separate "Author Comment". This will be the revised manuscript with tracked changes upon comments from all referees.**

References:

Baron, P. A. and KlausWilleke: Aerosol Measurement: Principles,Techniques, and Applications, 2 ed., edited by: Sons, J., Wiley-Interscience, 2001.

Hinds, W. C.: Aerosol Technology: Properties, Behavior, and Measurement of Airborne Particles, 2 ed., Wiley-Interscience, 1999.

Kiselev, A., Bachmann, F., Pedevilla, P., Cox, S. J., Michaelides, A., Gerthsen, D., and Leisner, T.: Active sites in heterogeneous ice nucleation—the example of K-rich feldspars, Science, doi: 10.1126/science.aai8034, 2016.

Libbrecht, K.: Growth rates of the principal facets of ice between −10°C and −40°C, Journal of Crystal Growth, 247, 530-540, doi: https://doi.org/10.1016/S0022-0248(02)01996-6, 2003.

Libbrecht, K. G.: The physics of snow crystals, Rep. Prog. Phys, 68, 855, doi: 10.1088/0034-4885/68/4/R03, 2005.

Ong, S., Zhao, X., and Eisenthal, K. B.: Polarization of water molecules at a charged interface: second harmonic studies of the silica/water interface, Chem. Phys. Lett., 191, 327-335, doi: 10.1016/0009-2614(92)85309-X, 1992.

Willeke, K. and Baron, P.: Aerosol Measurement: Principles, Techniques, and Applications, edited by: Klaus Willeke, P. A. B., Van Nostrand Reinhold, 1993.

Zhao, H., Bhattacharjee, S., Chow, R., Wallace, D., Masliyah, J. H., and Xu, Z.: Probing Surface Charge Potentials of Clay Basal Planes and Edges by Direct Force Measurements, Langmuir, 24, 12899-12910, doi: 10.1021/la802112h, 2008.

Zhao, X., Ong, S., and Eisenthal, K. B.: Polarization of water molecules at a charged interface. Second harmonic studies of charged monolayers at the air/water interface, Chem. Phys. Lett., 202, 513-520, doi: 10.1016/0009-2614(93)90041-X, 1993.

---

## Author Comment (AC3) · 13 Jun 2017

*Point-to-point answers to the comments of Referee #2*

*The author would like to thank the referee for his time reading the manuscript and for the recommendations and suggestions.*

RC: (1) The author stated that "the SHG signal is originated from the nonresonant OH stretching vibrations at the interface". (Line 8 on page 4) This statement does not make sense. A nonresonant signal is by definition not to be associated with a particular vibrational mode.

*AC: Indeed the statement "the SHG signal is originated from the nonresonant OH stretching vibrations at the interface", in page 2 line 19, is incorrect. The author has clarified this oversight failure in an earlier quick Author Comment on the discussion form because he expected that it may influence the general evaluation of the referees and the readers. This wording failure should however become clear to the reader when reaching to page 6 line 9 when the author started discussing interpretations in terms of "electric dipolar contribution" as an origin of the signal.*

*This oversight will be corrected in the revised version.*

RC: (2) Mica is birefringent. As the laser beam goes through the mica, its polarization may not be linear any more. "The advantage of using SM polarization combination is its dependence on only one non-vanishing nonlinear susceptibility tensor element (line 33 on page 3)" is likely not what has happened in the experiment.

*AC: Before starting the measurements, the polarization of the water SHG signal was analyzed and found to have the expected maxima at S and P polarizations. (Will be explained in the revised version)*

RC: (3) The index-matching gel of unknown chemical composition is a concern. While the gel helps to obtain the TIR condition, the gel/mica and gel/sapphire interfaces may produce SHG signal. The author may want to look into the freezing temperature of the gel too.

*AC: The author would like to thank the referee for pointing out this missing information.*

*The IMG was obtained from Thorlabs (source and specifications will be mentioned in the revised version). The Flash point is > 220 °C. However, the freezing point is not specified by the manufacturer but tested in the lab. At least the gel was not frozen until -45 °C.*

*As already mentioned in the original manuscript, P. 3, L. 11 – 13, the influence of the temperature dependence of the optical properties of the index-matching gel, sapphire prism and mica on the SHG signal was studied before starting the series of measurements. There was no significant effect of the changes in the optical properties of these media on the detected signal within the range of freezing temperatures observed in the study. The Figure below (Fig. R2.1) shows the SHG at mica-$N_2$ gas interface in the range of freezing temperatures mentioned in the manuscript.*

[Figure]

*Figure R2.1: SHG at mica-$N_2$ gas interface in the range of freezing temperatures observed in the study. The cell, shown in Figure S2 in the original SI, was filled with ultrapure $N_2$ gas (99.9999 %).*

*Figure R2.1 and the corresponding text will be added to the SI of the revised manuscript.*

(4) Figure 1 and 2 should be real-time plots similar to Figure 3.

*During the experiments, except for those of Figure 3, the signal was collected, averaged and stored as a function of the state of condensation, deposition, freezing, or formation of bulk. There was no need to record the data as a function of time. However, Fig. 3 is a special case because it was intended to report on the detected transient ice formed upon immersion freezing. Anyways, in the Figure below (Fig. R2.2), I show some screen shots of typical SHG signal and temperature changes in time, and show the points where the signal was collected and averaged.*

[Figure]

*Figure R2.2: Screen shots of typical signal and temperature changes in time during arbitrary cooling processes. The white line shows raw SHG data. The short red dashes below the SHG data show the points where the signal was collected and averaged. The red and green lines show the temperatures of the sample bottom and sample top respectively.*

*Figure R2.2 shows three examples of three arbitrary experiments (Runs):*

*Run1: The cell was purged continuously with $N_2$ gas while cooling the sample down to -40. There is a small gradual change in the signal due to the changes in the temperature dependent optical constants. The cell was then purged with humid air which resulted in a formation of an ice-film by deposition. Finally, the cell was purged again with $N_2$ gas which resulted in sublimation of the ice film.*

*Run2: The cell was purged with humid air (RH=5%) while cooling the sample down until the formation of an ice film and then the cell was purged by $N_2$ gas which resulted in film sublimation.*

*Run3: The cell was purged with humid air (RH=5%) while cooling the sample down until the formation of an ice film. Purging the cell with humid air was continued until the formation of bulk ice.*

*All experiments of deposition freezing were done in a similar way as described above. Figures 1 and 2 in the manuscript show the averaged signal at each event and the error bars on the figures show the fluctuation in the data points among all repeated experiments.*

*This part will be included in the revised manuscript.*

(5) Terms such as "liquid (film)", "liquid (bulk)", "transient ice", "stable ice" used in Figure 2 should be experimentally defined.

*This is a very good point and was missing in the original manuscript. The border between a film at the solid-air interface and a bulk (ice or water) was defined experimentally by the point where the intensity of a TIR reflected light from the solid-air interface drops due to the violation of TIR condition when the refractive index of the contact medium changes drastically from that of air ($n_a$=1) to one of those of water or ice ($n_{w \, or \, i}$ > 1.3). Whether the contact medium is ice or liquid, this was defined by the strong light scattering, observed at a CCD camera placed close to the detection path (Fig. R2.3), when the ice is formed. This camera was missing in Fig. S1 in the SI of the original manuscript and will be added in the revised version.*

*After immersion freezing, there was a rapid increase in the signal and then slow decrease. The maximum after the rapid increase was defined as the "transient ice" data point. After reaching a maximum, the signal was decreasing with time until stabilized after certain time. This stabilized signal was defined as the "stable ice" data point*

[Figure]

**Figure R3.3: The same as Fig. S1 but with including the CCD camera (Guppy F-036 Allied Vision Technology with LINOS Macro-CCD Lens 0.14x (1:7) f4)**

*The text in the manuscript and SI will be changed correspondingly.*

(6) The cooling rate dependence should be investigated.

*The results and conclusion were limited to the given cooling rate. Investigation of the influence of cooling rate was not planned in this work. However, the cooling rate dependence was tested in immersion freezing on sapphire surface using SFG in different study (under publication). It was found that the different cooling rates do not change the spectroscopic results.*

(7) One fundamental issue of SHG is that SHG intensity can be difficult to interpret. For example, why is there a _80% SHG drop from the "air" to the "bulk liquid" in Figure 2? What does the SHG measure?

*This is indeed a good question. The SHG signal is mainly produced by all polarizable species within the SHG-active region as long as the inversion symmetry is broken. The polarizable species at the surface are the water molecules and the surface OHs. The contribution of water molecules are limited by the penetration depth. The experiments were carried out in total internal reflection geometry with an incident angle of 15° from air to the prism side\*. The SHG-active region is limited by the penetration depth of the evanescent wave in the second medium (air, liquid water or ice). The calculated penetration depths are about 130 nm, 328 nm, and 253 nm for air, liquid water bulk, and ice bulk as contact media respectively. (This part will be added to the discussion in the revised version).*

*\* Note: there was a typo in the corresponding incidence angles at the interfaces in the original manuscript. This will be corrected in the revised version.*

*The 80% drop of the signal was explained in the final paragraph of page 6 in the original manuscript. However, the author agrees with the referee that this part was not well explained in the manuscript. Here is a more detailed discussion:*

*"Since SHG response reflects the overall arrangements of the polar entities at the interface between two isotropic media (Fordyce et al., 2001; Goh et al., 1988; Luca et al., 1995a), signal intensity is expected to increase when a single (or few non-centrosymmetric) layer(s) of water or ice is(are) formed at the surface. It is clear from Figures 1 and 2, however, that SHG intensity decreases upon deposition freezing, condensation, and growth of liquid layers by diffusion. This can be explained by phase interference between two signals originating from two different interfacial groups of opposite dipole moments: surface-OH points out of the surface and water points to the surface. A cleaved mica surface exhibits a disordered hexagonal arrangement of Si- (partly Al-) doubly coordinated O atoms in the outermost layer (Ostendorf et al., 2008). As an aluminosilicate mineral, this surface may protonate immediately in contact with the ambient air (forming silanol and aluminol groups at the surface). In this case, the mica-dry air signal can mainly originates from the surface hydroxyl groups (dangling−OH) which are naturally pointing out from the surface. Since mica surface is inherently negatively charged, the interfacial water will point to the surface and thus have a phase opposite to that of the surface−OH groups, as was reported by Shen and co-workers (Zhang et al., 2008) on negatively charged surfaces. This well explains the ostensible decrease in the overall signal upon deposition and condensation. When the surface of mica is covered with bulk water of a pH (∼ 7) relatively higher than the point-of-zero-charge (pzc) of the surface, it becomes totally deprotonated (no free−OH). Under this condition the SHG signal comes exclusively from the interfacial water molecules between the surface and liquid-bulk. This defines a new reference of the signal generated at the new interface (liquid-solid rather than gas-solid). The increase of the signal afterwards indicates more structuring of either the interfacial water before freezing, like shown in (Abdelmonem*

*et al., 2015) or the interfacial ice after freezing like shown here. It is worth to mention that, the presence of free surface hydroxyl groups on cleaved mica is poorly discussed in literature. Miranda et al. have indirectly referred to their (hydroxyl groups) existence in their work on mica-water vapor interface using SFG (Miranda et al., 1998). They used deuterated water ($D_2O$) in the SFG experiments to avoid confusion of the hydroxyl stretch modes in the spectrum from both water and mica. Maslova et al. have discussed it a bit explicit in their work on surface properties of cleaved mica and they assumed that hydroxyl groups of the basal plane are not reactive (Maslova et al., 2004). One in four Si in the tetrahedral layers of mica is randomly substituted for Al (Christenson and Thomson, 2016). The result is that the majority of the surface hydroxyl groups of a cleaved basal plane are silanol and therefore the surface deprotonation is mostly determined by silica. The pzc value of silica lies between its respective isoelectric points (pH 2-3, (Hartley et al., 1997; Iler, 1979; Scales et al., 1992)) and hence the silanols in silica (Si-OH) deprotonated totally in the presence of neutral water."*

*The final paragraph in the manuscript will be changed correspondingly.*

(8) Some conclusions made in the manuscript are not well supported by the SHG data. For example, line 8 on page 4: "The coincidence of the SHG signals of the thin ice film formed in DF2 and DF3 indicates identical structuring of water on the surface in two step deposition freezing regardless of the onset temperature." SHG simply cannot provide the structural information of water. The same SHG intensity does not necessarily mean the same water structure.

*SHG is electric dipole forbidden in the centrosymmetric bulk media and therefore SHG has been widely used as the spectroscopic probe for molecular orientation, structure, and spectroscopy of different interfaces (Richmond et al., 1988; Goh et al., 1988; Goh and Eisenthal, 1989; Zhao et al., 1993; Conboy et al., 1994; Luca et al., 1995b; Eisenthal, 1993; Eisenthal, 1996; Antoine et al., 1997; Fordyce et al., 2001; Zhang et al., 2005; Pham et al., 2017). The SHG response relates to the overall arrangements of the water (or more general interfacial) entities (Goh et al., 1988; Luca et al., 1995b; Fordyce et al., 2001). However, probably the sentence "… indicates identical structuring…" is overstated. This sentence will be tempered in the revised manuscript.*

*Finally the author would like to thank the referee again for his valuable comments which helped to improve the manuscript. The author would also like to emphasize that the aim of the work was to demonstrate the worthwhile of investigating different ice nucleation processes and water structuring upon freezing on the molecular-level using SHG spectroscopy. The manuscript is considered as a cornerstone of future complementary studies involving other surfaces, and other techniques to precisely investigate the layer thickness, the surface morphology effect, the cooling rates ...etc. Probably a better title of the manuscript could be "Direct molecular level characterization of different heterogeneous freezing modes on mica – Part 1"*

**NOTE: The modified text will be posted in a separate "Author Comment". This will be the revised manuscript with tracked changes upon comments from all referees.**

References:

Abdelmonem, A., Lützenkirchen, J., and Leisner, T.: Probing ice-nucleation processes on the molecular level using second harmonic generation spectroscopy, Atmos. Meas. Tech., 8, 3519-3526, doi: 10.5194/amt-8-3519-2015, 2015.

Antoine, R., Bianchi, F., F. Brevet, P., and H. Girault, H.: Studies of water/alcohol and air/alcohol interfaces by second harmonic generation, Journal of the Chemical Society, Faraday Transactions, 93, 3833-3838, doi: 10.1039/a703711b, 1997.

Conboy, J., Daschbach, J., and Richmond, G.: Studies of Alkane/Water Interfaces by Total Internal Reflection Second Harmonic Generation, The Journal of Physical Chemistry, 98, 9688-9692, doi: 10.1021/j100090a600, 1994.

Eisenthal, K. B.: Liquid interfaces, Accounts of Chemical Research, 26, 636-643, doi: 10.1021/ar00036a005, 1993.

Eisenthal, K. B.: Liquid Interfaces Probed by Second-Harmonic and Sum-Frequency Spectroscopy, Chem. Rev., 96, 1343-1360, doi: 10.1021/cr9502211, 1996.

Fordyce, A. J., Bullock, W. J., Timson, A. J., Haslam, S., Spencer-Smith, R. D., Alexander, A., and Frey, J. G.: The temperature dependence of surface second-harmonic generation from the air-water interface, Mol. Phys., 99, 677-687, doi: 10.1080/00268970010030022, 2001.

Goh, M. C., Hicks, J. M., Kemnitz, K., Pinto, G. R., Heinz, T. F., Eisenthal, K. B., and Bhattacharyya, K.: Absolute orientation of water molecules at the neat water surface, J. Phys. Chem., 92, 5074-5075, doi: 10.1021/j100329a003, 1988.

Goh, M. C. and Eisenthal, K. B.: The energetics of orientation at the liquid-vapor interface of water, Chemical Physics Letters, 157, 101-104, doi: 10.1016/0009-2614(89)87216-1, 1989.

Luca, A. A. T., Hebert, P., Brevet, P. F., and Girault, H. H.: Surface second-harmonic generation at air/solvent and solvent/solvent interfaces, J. Chem. Soc. Faraday Trans., 91, 1763-1768, doi: 10.1039/ft9959101763, 1995a.

Luca, A. A. T., Hebert, P., Brevet, P. F., and Girault, H. H.: Surface second-harmonic generation at air/solvent and solvent/solvent interfaces, Journal of the Chemical Society, Faraday Transactions, 91, 1763-1768, doi: 10.1039/ft9959101763, 1995b.

Ostendorf, F., Schmitz, C., Hirth, S., Kühnle, A., Kolodziej, J. J., and Reichling, M.: How flat is an air-cleaved mica surface?, Nanotechnology, 19, 305705, doi: 10.1088/0957-4484/19/30/305705, 2008.

Pham, T. T., Jonchère, A., Dufrêche, J.-F., Brevet, P.-F., and Diat, O.: Analysis of the second harmonic generation signal from a liquid/air and liquid/liquid interface, The Journal of Chemical Physics, 146, 144701, doi: 10.1063/1.4979879, 2017.

Richmond, G. L., Robinson, J. M., and Shannon, V. L.: Second harmonic generation studies of interfacial structure and dynamics, Prog. Surf. Sci., 28, 1-70, doi: 10.1016/0079-6816(88)90005-6, 1988.

Zhang, L., Tian, C., Waychunas, G. A., and Shen, Y. R.: Structures and Charging of α-Alumina (0001)/Water Interfaces Studied by Sum-Frequency Vibrational Spectroscopy, J. Am. Chem. Soc., 130, 7686-7694, doi: 10.1021/ja8011116, 2008.

Zhang, W.-k., Zheng, D.-s., Xu, Y.-y., Bian, H.-t., Guo, Y., and Wang, H.-f.: Reconsideration of second-harmonic generation from isotropic liquid interface: Broken Kleinman symmetry of neat air/water interface from dipolar contribution, The Journal of Chemical Physics, 123, 22471301 - 22471311, doi: 10.1063/1.2136875, 2005.

Zhao, X., Ong, S., and Eisenthal, K. B.: Polarization of water molecules at a charged interface. Second harmonic studies of charged monolayers at the air/water interface, Chemical Physics Letters, 202, 513-520, doi: 10.1016/0009-2614(93)90041-X, 1993.

---

## Author Comment (AC4) · 13 Jun 2017

*Point-to-point answers to the comments of SC #1*

*The author is grateful to Dr. Hugo K Christenson for his interest in the work and also for his valuable comments and constructive recommendations.*

SC: 1. The two-step nucleation process postulated by Campbell et al. in 2013 involved various organic liquids crystallising from vapour on mica surfaces, but only in surface features such as cleavage steps, cracks and pockets. A later study (Campbell et al., 2017)) has confirmed the two-step process for the organic liquids, and strongly suggested a two-step process for water and ice, although conclusive proof could not be obtained for water and ice.

*AC: Indeed, the discussion on the two-step process by (Campbell et al., 2013) in page 2 line 6 requires more details about the work. This will be considered in the revised version of the manuscript. The author would also like to thank Dr. Christenson for drawing the attention to the very recent work by Campbell et al. on the two-step process (Campbell et al., 2017). This work will also be considered in the revised manuscript.*

SC: 2. The material used by Layton and Harris in their 1963 paper (the reference is incomplete) was not muscovite mica but synthetic fluorophlogopite, which is similar in structure but has the hydroxyl groups replaced by fluorine. Moreover, neither is a metal oxide as stated in the abstract, but they are both layered aluminosilicates.

*AC: Thanks for this clarification. The difference between the two micas as well as the corrected citation will be included in the revised manuscript.*

SC: 3. The basal (cleavage) plane of mica consists of distorted hexagons of oxygen atoms, and there are no Si or Al atoms in the outermost layer. Moreover, the hydroxyl groups are below the oxygen hexagons, so there are none available for hydrogen bonding to adsorbed water molecules. This is explained in a recent review (Christenson and Thomson, 2016) which gives numerous references to the original literature on the structure determination of muscovite mica by X-ray diffraction.

*AC: Probably the sentence in the manuscript was not clear enough (will be revised). Indeed, a cleaved mica surface exhibits a distorted hexagons of oxygen atoms in the outermost layer (Ostendorf et al., 2008) which are doubly coordinated with Si/Al atoms. My hypothesis is as follow: As an aluminosilicate mineral, this surface may protonate immediately in contact with the ambient air (forming silanol and aluminol groups at the surface). In this case, the mica-dry air SHG signal can mainly originate from the surface hydroxyl groups (dangling –OH) which are naturally pointing out from the surface. A strong SHG signal was detected from the mica-air interface at 110 °C and under purging of $N_2$ gas before starting the cooling. The bulk hydroxyl groups cannot contribute to this signal. To my knowledge, the presence of free surface hydroxyl groups on mica is poorly discussed in literature. I could find only two articles discussing hydroxyl groups on mica basal plane: 1) Miranda et al. have indirectly referred to their (hydroxyl groups) existence in their work on mica-water vapor interface using SFG (Miranda et al., 1998). They wrote: "Deuterated water ($D_2O$) was used in the SFG*

*experiments to avoid confusion of the hydroxyl stretch modes in the spectrum from both water and mica". This should mean that they observed a signal from those species (surface OH) in the OH vibrational region and, for this reason, they moved to the OD vibrational region. 2) Maslova et al. have discussed it a bit explicit in their work on surface properties of cleaved mica and they assumed that hydroxyl groups of the basal plane are not reactive (Maslova et al., 2004).*

*Since one in four Si in the tetrahedral layers is randomly substituted for Al (Christenson and Thomson, 2016), the majority of the surface hydroxyl groups of a cleaved basal plane are silanol and therefore the surface deprotonation is mostly determined by silica. The point-of-zero-charge (pzc) value of silica lies between its respective isoelectric points (pH 2-3, (Hartley et al., 1997; Iler, 1979; Scales et al., 1992)) and hence the silanols in silica (Si-OH) deprotonated totally in the presence of neutral water.*

SC: 4. The simulation by Odelius et al. (1997) suggested that the mica surface is covered by a network of hydrogen bonds between water molecules only, with no free water hydroxyls. However, these results have been called into doubt by more recent simulations (Wang et al., 2005; Malani and Ayappa, 2009), and in a density-functional study that found no evidence of 2D-ice on mica, but that the properties of water on the surface are dominated by hydration of potassium ions (Feibelman 2013).

*AC: The author is thankful for this clarification and probably referring to this work in the last paragraph was not correct (will be removed in the revised version).*

SC: 5. The study of air-cleaved mica and water vapour (as is the case here for the measurements in water vapour) is complicated by the fact that the surface potassium ions are free to diffuse along the surface (as potassium carbonate), which may even result in the formation of crystallites on the surface in dry conditions. This was first shown by Christenson and Israelachvili in 1987, and was discussed in Balmer et al., 2008. The implications for studying mica in humid atmospheres was summarised in the 2016 review mentioned under point 3. Mica immersed in bulk water does not suffer from these problems, of course, as any potassium carbonate dissolves and the nature of the surface ions is determined by the pH and any residual electrolyte in the solution. The mobility of the potassium ions (as potassium carbonate) does not necessarily alter dramatically any measurement of average surface properties, as is the case with SHG, and at high humidities the potassium will be widely dispersed across the surface. However, the surface mobility of the potassium will necessarily decrease at lower temperatures, and there may be a dependence on the history of the mica surfaces. To summarise, it should be borne in mind that what is adsorbed to the mica surface is a thin film of aqueous potassium carbonate, the concentration of which varies with humidity and temperature, rather than pure water.

*AC: As written in the original manuscript, P. 3, L. 23, the laser power was reduced to allow observable signal but not destroy the index-matching gel. The fluctuation in the signal due to reducing the laser power limited the sensitivity of the system to monitor minor changes which may arise from pre-adsorption of sub-monolayers or even few monolayers at temperatures higher than the dew point or freezing point. The thickness of the water layer at the time of collecting the signal is expected to be in the order of 1 µm (will be explained in the revised version) which should allow the potassium to be*

*widely dispersed across the surface. In addition, the sample was washed several times in the cell and kept in either dry $N_2$ gas or humid air. The humid air was obtained from continuously $N_2$-purged MQ water as described in the SI. It is also expected, under these conditions, that the $K^+$ ions are totally replaced by $H^+$ ions. Studying monolayer film needs experimental arrangements which are not available in our Lab at this moment.*

*NOTE: The modified text will be posted in a separate "Author Comment". This will be the revised manuscript with tracked changes upon comments from all referees.*

References:

Campbell, J. M., Meldrum, F. C., and Christenson, H. K.: Characterization of Preferred Crystal Nucleation Sites on Mica Surfaces, Cryst. Growth Des., 13, 1915-1925, doi: 10.1021/cg301715n, 2013.

Campbell, J. M., Meldrum, F. C., and Christenson, H. K.: Observing the formation of ice and organic crystals in active sites, Proceedings of the National Academy of Sciences, 114, 810-815, doi: 10.1073/pnas.1617717114, 2017.

Christenson, H. K. and Thomson, N. H.: The nature of the air-cleaved mica surface, Surface Science Reports, 71, 367-390, doi: https://doi.org/10.1016/j.surfrep.2016.03.001, 2016.

Hartley, P. G., Larson, I., and Scales, P. J.: Electrokinetic and Direct Force Measurements between Silica and Mica Surfaces in Dilute Electrolyte Solutions, Langmuir, 13, 2207-2214, doi: 10.1021/la960997c, 1997.

Iler, R. K.: The Chemistry of Silica: Solubility, Polymerization, Colloid and Surface Properties and Biochemistry of Silica, John Wiley and Sons, New York, 896 pp., 1979.

Maslova, M. V., Gerasimova, L. G., and Forsling, W.: Surface Properties of Cleaved Mica, Colloid Journal, 66, 322-328, doi: 10.1023/B:COLL.0000030843.30563.c9, 2004.

Miranda, P. B., Xu, L., Shen, Y. R., and Salmeron, M.: Icelike Water Monolayer Adsorbed on Mica at Room Temperature, Physical Review Letters, 81, 5876-5879, doi: 10.1103/PhysRevLett.81.5876, 1998.

Ostendorf, F., Schmitz, C., Hirth, S., Kühnle, A., Kolodziej, J. J., and Reichling, M.: How flat is an air-cleaved mica surface?, Nanotechnology, 19, 305705, doi: 10.1088/0957-4484/19/30/305705, 2008.

Scales, P. J., Grieser, F., Healy, T. W., White, L. R., and Chan, D. Y. C.: Electrokinetics of the silica-solution interface: a flat plate streaming potential study, Langmuir, 8, 965-974, doi: 10.1021/la00039a037, 1992.

---

## Author Comment (AC5)

[revised manuscript text omitted]

-   *Data acquisition*

During the SHG experiments the signal was collected, averaged and stored as a function of the state
of condensation, deposition, freezing, or formation of bulk. These are the data points plotted in Figs.
3 and 4 in the manuscript. Only for Fig. 5 the data was recorded as a function of time to report on the
detected transient ice formed upon immersion freezing. Figure S2, shows some screen shots of
typical SHG signal and temperature changes in time, and shows the points where the signal was
collected and averaged for Figs. 3 and 4.

[Figure]

**Figure S2: Screen shots of typical signal and temperature changes in time during arbitrary cooling processes. The white**
**line shows raw SHG data. The short red dashes below the SHG data show the points where the signal was collected and**
**averaged. The red and green lines show the temperatures of the sample bottom and sample top respectively.**

Figure S2 shows three examples of three arbitrary runs performed on sapphire (only for
demonstration purpose):

Run1: The cell was purged continuously with $N_2$ gas while cooling the sample down to -40. There is a
small gradual change in the signal due to the changes in the temperature dependent optical
constants. The cell was then purged with humid air which resulted in a formation of an ice-film by
deposition. Finally, the cell was purged again with $N_2$ gas which resulted in sublimation of the ice
film.

Run2: The cell was purged with humid air (RH=5%) while cooling the sample down until the
formation of an ice film and then the cell was purged by $N_2$ gas which resulted in film sublimation.

Run3: The cell was purged with humid air (RH=5%) while cooling the sample down until the formation of an ice film. Purging the cell with humid air was continued until the formation of bulk ice.

All experiments of deposition freezing were done in a similar way as described above. Figs. 3 and 4 in the manuscript show the averaged signal at each event and the error bars on the figures show the fluctuation in the data points among all repeated experiments.

*- Fresnel factors*

The SHG signal with a frequency $\omega_{SH} = 2\omega_{in}$ under SM polarization, (S-polarized SHG and 45°-polarized incident light), generated from an incoming visible light with a frequency $\omega_{in}$ can be described with the equation:

$$S_{SM}(\omega_{SH}) \propto \left| L_{yy}(\omega_{SH}) L_{yy}(\omega_{in}) L_{zz}(\omega_{in}) \chi_{yyz}^{(2)} \right|^2 I_{in}^2 \tag{2}$$

where, $L(\omega_i)$ is the nonlinear Fresnel factor at $\omega_i$ and $\chi_{yyz}^{(2)}$ is the surface second order nonlinear susceptibility tensor for SM polarisation combination (equivalent to SSP in SFG) (Shen, 1989b; Zhuang et al., 1999) and a measure of the degree of molecular ordering. To obtain the molecular quantity $\chi_{yyz}^{(2)}$ the measured SHG intensities have thus to be normalized to the Fresnel factors which are optical constants dependent. Figure S3 shows the change of Fresnel factors as a function of refractive index in the range of refractive indices covering the involved media in this work. The SHG intensities reported in Figures 3 and 4 in the manuscript are Fresnel corrected and thus directly proportional to $\left| \chi_{yyz}^{(2)} \right|^{(2)}$.

[Figure]

**Figure S3: Theoretically calculated Fresnel factors for sapphire-water (red line) and mica-water (green line) interfaces at incident angle of 15° from air to the sapphire prism. Optical geometry can be found in (Abdelmonem et al., 2015; Abdelmonem et al., 2017).**

The Fresnel factors are expected to change with temperature and hence affect the measured SHG
signal. The influence of the temperature dependence of the optical properties of the IMG, sapphire
prism and mica on the SHG signal was studied before starting the series of measurements. Figure S4
shows the SHG at mica-$N_2$ gas interface in the mentioned range of temperatures. There was no
significant effect of the changes in the optical properties of these media on the detected signal
within the range of freezing temperatures observed in the study.

[Figure]

**Figure S4: SHG at mica-$N_2$ gas interface in the range of freezing temperatures observed in the study. The cell, shown in Figure 2, was filled with ultrapure $N_2$ gas (99.9999 %).**

---

## Author Response (AR2)

**Direct molecular level characterization of different heterogeneous freezing modes on mica – Part 1**

**Ahmed Abdelmonem[1]**

**[1]Institute of Meteorology and Climate Research – Atmospheric Aerosol Research (IMKAAF), Karlsruhe Institute of Technology (KIT), 76344 Eggenstein-Leopoldshafen, Germany**

*Correspondence to*: **Ahmed Abdelmonem (ahmed.abdelmonem@kit.edu)**

**Final Rebuttal**

**2nd Review**

**I. Point-to-point response to Reviewers**

**II. Revised manuscript with tracked changes**

**I. Point-to-point response to Reviewers**

Reviewer #1:

RC: The author refers to surface hydroxyl groups on mica (p9, l10-20) although there are none. This is an important point, because it has a bearing on the on the discussion of the orientation of water molecules at the surface. Mica does not protonate by forming silanol groups, the effect of water at moderate or low pH is to replace the potassium ions with hydrogen ions. The author should consult the extensive literature on mica again, for although some is cited, the information has not been correctly reported.

*AC: I am grateful for the referee to have pointed out the issue about the hydroxyls on mica. After consulting colleagues at KIT, I am now rather convinced that there are no hydroxyls on the mica basal plane, if one starts from the crystal structure. What happens in the presence of water (even during sample preparation in normal lab-atmosphere) is another issue. However, the similar behaviour of mica and sapphire basal planes actually supports the idea that the structural hydroxyls (on sapphire basal planes they are known to be present) should not play a role in the interpretation of the results. Rather it would make sense that the initial signal comes from a water film that is present on both surfaces and may be quite similar. Subsequently, water interacts with this water film and this causes the observations.*

*The final paragraph has been significantly changed in the view of this perspective to:*

*"Finally, an explanation is warranted why there is a drop, rather than an increase, of the signal upon adsorption of water (or ice) on the surface (of either mica or sapphire). The SHG response reflects the overall arrangements of the polar entities at the interface between two isotropic media (Fordyce et al., 2001; Goh et al., 1988; Luca et al., 1995). This signal intensity is expected to increase when a single (or few non-centrosymmetric) layer(s) of water or ice is(are) formed at (or added to) the surface, while SHG intensity decreases upon deposition freezing, condensation, and growth of liquid layers by diffusion as can be seen in Figs. 3 and 4. One expectation could be phase interference between two signals originating from two different interfacial entities of opposite dipole moments. The possible contributors could be surface hydroxyls, pre-adsorbed water (films) and the added water. The surface hydroxyls can probably be ruled out, since mica-basal planes based on structural considerations should not have such groups, while sapphire-basal planes do. Thus pre-adsorbed water would be the more reasonable option. Actually on sapphire basal planes such water films have been proposed based on experimental data (Lützenkirchen et al., 2010) and reported in MD-simulations (Argyris et al., 2011). This scenario which may explain the decrease in SHG intensity upon deposition freezing, condensation, and growth of liquid layers involves the pre-existence of 2D bilayer of water on the surface. This 2D bilayer has an ordered structure of the water monolayer, which greatly enhances the numbers of H bonds inside the monolayer. This in turn reduces the likelihood of H-bond formation between the water molecules of this monolayer with other molecules, a phenomenon which makes hydrophilic surface hydrophobic (Wang et al., 2009; Hu and Michaelides, 2007; Ranea et al., 2009). Adjacent layers would then be comparable to those on hydrophobic surfaces and could yield the signal interference that is observed. Clearly, the drop in signal merits further investigation, in particular since pre-adsorbed water films are not necessarily identical for mica and sapphire. In particular the mica surface exhibits a quite complex interfacial hydration structure (Cheng et al., 2001)."*

*The paragraph in page 9 is modified in accordance.*

Argyris, D., Ho, T., Cole, D. R., and Striolo, A.: Molecular Dynamics Studies of Interfacial Water at the Alumina Surface, J. Phys. Chem. C, 115, 2038-2046, doi: 10.1021/jp109244c, 2011.

Cheng, L., Fenter, P., Nagy, K. L., Schlegel, M. L., and Sturchio, N. C.: Molecular-Scale Density Oscillations in Water Adjacent to a Mica Surface, Phys. Rev. Lett., 87, 156103, 2001.

Fordyce, A. J., Bullock, W. J., Timson, A. J., Haslam, S., Spencer-Smith, R. D., Alexander, A., and Frey, J. G.: The temperature dependence of surface second-harmonic generation from the air-water interface, Mol. Phys., 99, 677-687, doi: 10.1080/00268970010030022, 2001.

Goh, M. C., Hicks, J. M., Kemnitz, K., Pinto, G. R., Heinz, T. F., Eisenthal, K. B., and Bhattacharyya, K.: Absolute orientation of water molecules at the neat water surface, J. Phys. Chem., 92, 5074-5075, doi: 10.1021/j100329a003, 1988.

Hu, X. L. and Michaelides, A.: Ice formation on kaolinite: Lattice match or amphoterism?, Surf. Sci., 601, 5378-5381, doi: 10.1016/j.susc.2007.09.012, 2007.

Luca, A. A. T., Hebert, P., Brevet, P. F., and Girault, H. H.: Surface second-harmonic generation at air/solvent and solvent/solvent interfaces, J. Chem. Soc. Faraday Trans., 91, 1763-1768, doi: 10.1039/ft9959101763, 1995.

Lützenkirchen, J., Zimmermann, R., Preočanin, T., Filby, A., Kupcik, T., Küttner, D., Abdelmonem, A., Schild, D., Rabung, T., Plaschke, M., Brandenstein, F., Werner, C., and Geckeis, H.: An attempt to explain bimodal behaviour of the sapphire c-plane electrolyte interface, Adv. Colloid Interface Sci., 157, 61-74, doi: 10.1016/j.cis.2010.03.003, 2010.

Ranea, V. A., Carmichael, I., and Schneider, W. F.: DFT Investigation of Intermediate Steps in the Hydrolysis of $\alpha$-Al2O3(0001), J. Phys. Chem. C, 113, 2149-2158, doi: 10.1021/jp8069892, 2009.

Wang, C., Lu, H., Wang, Z., Xiu, P., Zhou, B., Zuo, G., Wan, R., Hu, J., and Fang, H.: Stable Liquid Water Droplet on a Water Monolayer Formed at Room Temperature on Ionic Model Substrates, Phys. Rev. Lett., 103, 137801, 2009.

Reviewer #2:

(1) Page 1 in the point-to-point response to Referee #2.

AC: "Before starting the measurements, the polarization of the water SHG signal was analyzed and found to have the expected maxima at S and P polarizations. (Has been mentioned in the revised version, P. 3, L. 35-37)"

Reviewer 2: It is not clear what are "the expected maxima at S and P polarizations." Because mica is birefringent, the polarizations of the incident beam and the output SHG may be rotated. This effect is related to the incident plane, the incident angle, and the thickness of the individual mica used in the experiments.

*AC (1): Water molecule belongs to the $C_{2v}$ group. The SHG in terms of the incident and generated polarization has the relation shown in Fig. R2_1. There are maxima at "p-in − p-out" polarization combination and "45-in − s-out" polarization combination. The positions of these two maxima were cheeked for the signal of water in contact with mica. However, in case it happens, the rotation of polarization wouldn't significantly affect the results in terms of degree of ordering. As stated in page 6, line 3, the experiments were carried out at an angle (~ 63.4°) close to the critical angle of TIR (~59.2°) for mica-water interface. Under such conditions both polarization combinations give indication to the degree of ordering of water molecules.*

[Figure]

*Fig. R2_1. 360° polarization SHG measurement of the neat air/water interface at two fixed detection polarizations (open circle for p detection and closed circle for s detection). The lines are fitting curves. (FIG. 1 from (Zhang et al., 2005))*

(2) Page 4 in the point-to-point response to Referee #2.

AC: "The border between a film at the solid-air interface and a bulk (ice or water) was defined experimentally by the point where the intensity of a TIR **reflected** light from the solid-air interface drops due to the violation of TIR condition when the refractive index of the contact medium changes drastically from that of air (na=1) to one of those of water or ice (nw or i > 1.3)."

Reviewer 2: If this is the experimental setup, the SHG drop in Figure 4 is highly related to the Fresnel factors calculated by the author. As the Fresnel factors depend on the polarization of light, the birefringent property of mica has to be considered or ruled out. Data before and after the Fresnel factor correction should be shown in Figure 4.

*AC (2): The situation of the TIR SHG measurement was different and, as stated in first paragraph of page 6 in the manuscript, "the incident angle from air of the fundamental beam was adjusted to 15° with respect to surface normal of the outer side of the sapphire prism. The corresponding incident angle on the mica-air or -water interface was ~ 63.4°, which is __higher__ than the critical angles of TIR for mica-air (~39.7°) __and__ mica-water (~59.2°) interfaces. This guaranteed a TIR condition regardless of any changes in the Fresnel factors caused by the change of refractive indices with changing temperature". So, TIR condition was preserved for the SHG measurements. However, due to the large difference between the refractive index of air and those of water and ice, Fresnel corrections were made. The Fresnel factors have been already shown in Fig. S3 (= 3.5, 44.5 and 28.1 for air-, water-, and ice-mica interfaces respectively). Data before and after the Fresnel factor correction have not been shown in Figure 4 because the effect of Fresnel factor correction on data is not the focus of the paper. Figure R2_2 shows the data of Fig. 4 before and after Fresnel correction. These data are included now in the SI.*

[Figure]

*Figure R2_2. Data in Figure 4 of the manuscript before and after Fresnel correction. (a) Original data before Fresnel correction, (b) Data after Fresnel correction and (c) Data after Fresnel correction and normalization to the mica-air signal  (= Figure 4 in the manuscript)*

Zhang, W.-k., Zheng, D.-s., Xu, Y.-y., Bian, H.-t., Guo, Y., and Wang, H.-f.: Reconsideration of second-harmonic generation from isotropic liquid interface: Broken Kleinman symmetry of neat air/water interface from dipolar contribution, The Journal of Chemical Physics, 123, 22471301 - 22471311, doi: 10.1063/1.2136875, 2005.

**II. Revised manuscript with tracked changes**

[revised manuscript text omitted]

Ostendorf, F., Schmitz, C., Hirth, S., Kühnle, A., Kolodziej, J. J., and Reichling, M.: How flat is an air-cleaved mica surface?, Nanotechnology, 19, 305705, doi: 10.1088/0957-4484/19/30/305705, 2008.

Poppa, H. and Elliot, A. G.: The surface composition of Mica substrates, Surf. Sci., 24, 149-163, doi: 10.1016/0039-6028(71)90225-1, 1971.

Pruppacher, H. R. and Klett, J. D.: Microphysics of clouds and precipitation, 2 ed., Atmospheric and oceanographic sciences library, 18, Kluwer Academic Publishers, Dordrecht ; Boston, 954 pp., 1997.

Ranea, V. A., Carmichael, I., and Schneider, W. F.: DFT Investigation of Intermediate Steps in the Hydrolysis of α-Al2O3(0001), J. Phys. Chem. C, 113, 2149-2158, doi: 10.1021/jp8069892, 2009.

Rao, Y., Tao, Y.-s., and Wang, H.-f.: Quantitative analysis of orientational order in the molecular monolayer by surface second harmonic generation, J. Chem. Phys., 119, 5226-5236, doi: 10.1063/1.1597195, 2003.

Schaefer, V. J.: The Formation of Ice Crystals in the Laboratory and the Atmosphere, Chem. Rev., 44, 291-320, doi: 10.1021/cr60138a004, 1949.

Shen, Y. R.: Optical Second Harmonic Generation at Interfaces, Annu. Rev. Phys. Chem., 40, 327-350, doi: 10.1146/annurev.pc.40.100189.001551, 1989a.

Shen, Y. R.: Surface properties probed by second-harmonic and sum-frequency generation, Nature, 337, 519-525, doi: 10.1038/337519a0, 1989b.

Slater, B., Michaelides, A., Salzmann, C. G., and Lohmann, U.: A Blue-Sky Approach to Understanding Cloud Formation, Bull. Am. Meteorol. Soc., 97, 1797-1802, doi: 10.1175/bams-d-15-00131.1, 2016.

Wang, C., Lu, H., Wang, Z., Xiu, P., Zhou, B., Zuo, G., Wan, R., Hu, J., and Fang, H.: Stable Liquid Water Droplet on a Water Monolayer Formed at Room Temperature on Ionic Model Substrates, Phys. Rev. Lett., 103, 137801, 2009.

Willeke, K. and Baron, P.: Aerosol Measurement: Principles, Techniques, and Applications, edited by: Klaus Willeke, P. A. B., Van Nostrand Reinhold, 1993.

Zhang, L., Tian, C., Waychunas, G. A., and Shen, Y. R.: Structures and Charging of α-Alumina (0001)/Water Interfaces Studied by Sum-Frequency Vibrational Spectroscopy, J. Am. Chem. Soc., 130, 7686-7694, doi: 10.1021/ja8011116, 2008.

Zhang, Z. Z. and Bailey, G. W.: Reactivity of basal surfaces, steps and edges of muscovite; an AFM study, Clays Clay Miner., 46, 290-300, 1998.

Zhao, X., Ong, S., and Eisenthal, K. B.: Polarization of water molecules at a charged interface. Second harmonic studies of charged monolayers at the air/water interface, Chem. Phys. Lett., 202, 513-520, doi: 10.1016/0009-2614(93)90041-X, 1993.

Zhuang, X., Miranda, P. B., Kim, D., and Shen, Y. R.: Mapping molecular orientation and conformation at interfaces by surface nonlinear optics, Phys. Rev. B, 59, 12632-12640, doi: 10.1103/PhysRevB.59.12632, 1999.

---

## Author Response (AR3)

**Direct molecular level characterization of different heterogeneous freezing modes on mica – Part 1**

**Ahmed Abdelmonem[1]**

**[1]Institute of Meteorology and Climate Research – Atmospheric Aerosol Research (IMKAAF), Karlsruhe Institute of Technology (KIT), 76344 Eggenstein-Leopoldshafen, Germany**

*Correspondence to*: **Ahmed Abdelmonem (ahmed.abdelmonem@kit.edu)**

**Final Rebuttal**

**ACP**

**I. Point-to-point response to Co-editor**

**II. Revised manuscript with tracked changes**

**I. Point-to-point response to Co-editor**

Technical corrections:

EC: Page 2, line 5: What does "contains a water-soluble compound, almost potassium carbonate crystallites" mean? The word "almost" is distracting/confusing.
*AC: Indeed, the word "almost" is confusing (removed)*

EC: Page 2, line 40: I suggest the following change "The change of pH with temperature is known for neutral water (…)".
*AC: Done*

EC: Page 3, line 6: I suggest to discard "which will contribute to its rapid development in the next years."
*AC: Discarded*

EC: Page 5, line 20: Please omit "logical".
*AC: Omitted*

EC: Page 8, line 8: Omit "big".
*AC: Omitted*

EC: Page 10, l. 17: Correct "weak".
*AC: Corrected*

SI:
EC: Equation 1: I suggest to leave out the "dot" or place a "center dot" to indicate multiplication.
*AC: Dot replaced by center dot*

EC: Figure caption S4: Period missing.

*AC: Done*

**II. Revised manuscript with tracked changes**

[revised manuscript text omitted]